# Cell Cycle and DNA Repair Regulation in the Damage Response: Protein Phosphatases Take Over the Reins

**DOI:** 10.3390/ijms21020446

**Published:** 2020-01-10

**Authors:** Adrián Campos, Andrés Clemente-Blanco

**Affiliations:** Instituto de Biología Funcional y Genómica (IBFG). Consejo Superior de Investigaciones Científicas (CSIC) and Universidad de Salamanca (USAL), 37007 Salamanca, Spain; adrian.camposf@gmail.com

**Keywords:** phosphatases, cell cycle, DNA damage checkpoint, DNA repair, checkpoint recovery

## Abstract

Cells are constantly suffering genotoxic stresses that affect the integrity of our genetic material. Genotoxic insults must be repaired to avoid the loss or inappropriate transmission of the genetic information, a situation that could lead to the appearance of developmental abnormalities and tumorigenesis. To combat this threat, eukaryotic cells have evolved a set of sophisticated molecular mechanisms that are collectively known as the DNA damage response (DDR). This surveillance system controls several aspects of the cellular response, including the detection of lesions, a temporary cell cycle arrest, and the repair of the broken DNA. While the regulation of the DDR by numerous kinases has been well documented over the last decade, the complex roles of protein dephosphorylation have only recently begun to be investigated. Here, we review recent progress in the characterization of DDR-related protein phosphatases during the response to a DNA lesion, focusing mainly on their ability to modulate the DNA damage checkpoint and the repair of the damaged DNA. We also discuss their protein composition and structure, target specificity, and biochemical regulation along the different stages encompassed in the DDR. The compilation of this information will allow us to better comprehend the physiological significance of protein dephosphorylation in the maintenance of genome integrity and cell viability in response to genotoxic stress.

## 1. Introduction

The maintenance of genome integrity is an essential feature of cellular physiology. Our hereditary information encoded in DNA is intrinsically susceptible to alteration, being continually threatened by both endogenous and environmental genotoxic stresses that could alter our genomic material. Environmental sources of DNA damage include several DNA damaging agents, such as ionizing radiation (IR), ultraviolet radiation (UV), alkylating compounds, viral infections and chemotherapeutic drugs. On the other hand, multiple endogenous factors can also compromise genome integrity, such as reactive oxygen species that are generated during cell metabolism or unavoidable errors of certain cellular processes, such as DNA replication or meiotic recombination. These endogenous DNA damaging agents may cause hydrolysis, oxidation, alkylation or the mismatching of DNA bases, thus hazarding the stability of the genetic material [1]. It has been estimated that every cell of our body is exposed to up to 10^5^ of these DNA lesions per day. Most of these adducts might directly or indirectly undergo the formation of single-strand breaks (SSB) that arise when one of the two strands of the DNA duplex is severed. This relatively-easily-to-repair lesion might evolve into a more dangerous form of DNA damage, the double-strand break (DSB), where both DNA strands are simultaneously excised. To counter all these distinctive types of genetic alterations, cells have acquired specialized systems that detect the lesions, signal their presence and, at last, mediate their repair in order to safeguard the stability of our genome. The orchestration of these processes is tightly regulated by the activation of a highly evolutionarily conserved and sophisticated network that is generically termed as the DNA damage response (DDR). When the DDR fails or the rate of DNA damage exceeds the capacity of the pathway to cope with it, the increased accumulation of DNA alterations can lead to the appearance of mutations, thus subjecting cells to a higher risk of malignant transformation [2,3]. As a matter of fact, errors in the execution of this surveillance pathway have been linked to the development of various congenital human disorders [2,4,5].

At the molecular level, the DDR is a complex signaling pathway that is integrated by a large number of factors that actively contribute to the organization of the multiple cellular events that are enclosed in the response. Though most of these events are, in principle, independent processes, cells have to precisely coordinate them in space and time to guarantee the accurate restoration of the DNA molecule [6]. As with other canonical signaling pathways, the DDR is composed of sensors, transducers and effectors [7]. Sensors are able to recognize the presence of damage or aberrant structures in the DNA molecule and trigger a signal throughout the transducers to a set of effectors that participate in a broad range of cellular processes, such as DNA replication, DNA repair and cell cycle progression. Upon the generation of a DNA lesion, the DDR coordinates the response by activating two parallel routes: (1) the DNA damage checkpoint, a cell cycle blockage that restrains chromosome segregation until the damaged DNA has been fixed, and (2) the stimulation of a set of DNA repair factors that deal with the restoration of the broken DNA. The activation of the DNA damage checkpoint restrains the mitotic entry of cells containing DNA adducts by triggering a transduction pathway that ends up taking over the control of the cell cycle, thus blocking cells in G2/M. This DNA damage-dependent arrest allows for time for the repair machinery to act on the DNA lesion and fix it. To date, numerous repair mechanisms that operate in different stages of the cell cycle and under distinctive types of DNA lesions have been discovered. Formally, these DNA damage repair systems have been classified in the following subgroups: homologous recombination (HR) and non-homologous end joining (NHEJ), involved in the repair of DSBs; nucleotide excision repair (NER), a molecular pathway that removes bulky DNA lesions that are caused by exposure to UV light or environmental mutagens; base excision repair (BER), a mechanism that operates in the repair of DNA lesions that cause distortions in the DNA helix structure such as oxidation, deamination, and alkylation; and mismatch repair (MMR), a surveillance system that detects and corrects the base mispairing generated during the replication process [4,8]. After the DNA lesion has been repaired, the DNA damage checkpoint must be deactivated. This inactivation, most commonly known as checkpoint recovery, is required to exit from the G2/M arrest and resume cell cycle progression.

During the last few years, there has been an increasing interest in the molecular mechanisms behind the regulation of the DDR. In response to a DNA lesion, several DDR-specific kinases transmit the signal from the sensors to numerous biological processes across the cell by phosphorylating multiple DDR components at specific serine and threonine residues [9,10]. Importantly, phosphorylation events must be reversible to ensure the transient activation of the DNA damage response [9,10,11]. This reversibility of the system ensures a fine-tuning of the pathway in order to avoid the illicit activation of the damage response in the absence of DNA damage, as well as to allow for a fast cessation of the signal once the damaged DNA has been fixed. Supporting this notion, the capacity of cells to reverse protein phosphorylation by DDR-phosphatases is essential to endorse the tight, reversible, and adjustable control of the DDR [12]. Lately, knowledge of the structure and functions of multiple DDR-phosphatases has been gaining strength and interest in the field, and, although their in-depth mechanisms of action are not yet known in most cases, numerous studies have demonstrated that they are key players in the regulation of multiple stages of the damage response [13].

Protein phosphatases can be classified into different groups on the basis of their sequence, structure and catalytic mechanisms (Table 1). These groups comprise the classic serine/threonine phosphoprotein phosphatases (PPP) family (PP1, PP2A, PP2B, PP4, PP5, PP6 and PP7), the protein phosphatase Mg2+ or Mn2+ dependent (PPM) family (PP2C, WIP1) and the cyclin-dependent kinase (CDK)-antagonizing cell division cycle 14 (CDC14), a member of the dual-specificity phosphatase (DUSP) family. Curiously, it is remarkable to perceive a significant reduction in the total amount of phosphatases when compared to the number of protein kinases operating in the cell. For a long time, this observation led to the idea that protein phosphatases were promiscuous and non-specific enzymes. However, we now know that protein phosphatases are indeed highly specialized complexes that exhibit an extremely substrate specificity [14,15,16]. In most cases, this is attained by their ability to form stable complexes with a wide range of regulatory subunits that provide the essential determinants for subcellular localization, substrate specificity, and the fine-tuning of their phosphatase activity [17].

Recently, new evidences have demonstrated that protein phosphatases cooperate in the regulation of the DDR, not only by promoting cell cycle re-entry after the DNA lesion has been restored but also by playing important roles in the execution of different steps of the repair process. In this regard, it has been postulated that each phosphatase could work on different DNA structures during different steps of the repair process and in response to different types of DNA damage [18]. Supporting this perception, several DNA repair factors have multiple phospho-residues that are concurrently phosphorylated and dephosphorylated, and their kinetics are potentially regulated by the presence of DDR-kinase/phosphatase molecular switches [19,20,21,22,23]. Moreover, the activity of protein phosphatases is regulated by their temporal interaction with specific regulators, which makes their study even more appealing [24]. Finally, recent evidences have established a connection between the lack of activity of DDR-related phosphatases with the appearance of diseases in higher eukaryotes, mirroring the importance of protein dephosphorylation in the maintenance of genome integrity for health and development in humans.

In this review, we summarize recent discoveries about the role of DDR-phosphatases during the generation of a DNA adduct. We also focus our attention on the physiological significance of these enzymes during the repair of a DNA lesion, how their enzymatic activity is regulated, and what their targets are in the process. This compilation will allow us to acquire a complete picture of the importance of protein dephosphorylation during the execution of the DNA damage response. 

## 2. Protein Phosphatases in the Control of the DNA Damage Checkpoint

Checkpoints are signal transduction pathways that coordinate DNA damage with DNA repair and cell cycle progression. After a DNA breakage has been inflicted, the first priority of the cell is to detect it as quickly and efficiently as possible in order to elicit an accurate activation of the DNA damage checkpoint. The timely activation of this checkpoint restricts damaged cells to premature entry into mitosis and stimulates the activity of repair factors involved in the restoration of the DNA lesion. The DNA damage checkpoint is initiated by the recognition of the DNA lesion by the phosphatidyl inositol kinase-like kinase (PIKK) Mec1/ATR and Tel1/ATM [25] (Figure 1). The recruitment of Mec1/ATR and Tel1/ATM to the DNA lesion triggers a phosphorylation cascade through several transducer kinases, such as Rad53/CHK2 or Chk1/CHK1, in a process that relies on the support of several adaptor proteins, including Rad9/53BP1, Mrc1/Claspin and Ddc2/ATRIP (Figure 1). Even though the accurate initiation and maintenance of the DNA damage checkpoint relies on the precise activation of DNA damage-specific kinases [26], it has recently been demonstrated that the CDK also collaborates in the response by phosphorylating multiple targets of the pathway [27]. These kinases spread the signal to downstream targets involved in the execution of different biological processes that are encompassed in the DDR, including a transient G2/M arrest. This cell cycle block is mainly attained by overtaking the control of canonical CDK regulators, such as Cdc25/CDC25A-B or Swe1/WEE1 [28]. Hence, the activation of the DNA damage checkpoint leads to an increased activation of the CDK inhibitor WEE1 [29,30] and a degradation or inactivation of the CDK stimulators CDC25A [31] and CDC25B [32], respectively (Figure 1). This diminishes the levels of CDK activity, thus restraining cell cycle progression and avoiding entrance into mitosis as long as the DNA damage persists. In this section, we discuss the specific functions of each protein phosphatase in the regulation of the different substrates operating in the DNA damage checkpoint. We also focus on the consequences of disturbing phosphatase proficiency in the correct activation and maintenance of the DNA damage signal.

### 2.1. Protein Phosphatase 1

Though the activation of the DNA damage checkpoint is mainly driven by the phosphorylation events of its components, their steady state phosphorylation must be highly regulated by kinases and phosphatases that work in tandem. This kinase-phosphatase balance allows for the tight control of the response and prevents cells from an unnecessary activation of the signal in the absence of DNA damage. PP1 is one of the most active phosphatases in the cell and is responsible for the dephosphorylation of one third of all phospho-proteins [33]. It is composed of a catalytic subunit and multiple regulatory elements that provide specificity for a great variety of targets. The central region of the catalytic subunit is almost identical, in terms of amino-acid sequence, between different species. However, some species-specific differences are observed in the N- and C-terminal domains of the catalytic subunit [34]. The conserved region of PP1’s catalytic subunit is similar to that of PP2A, a feature that may explain the overlapping roles of both phosphatases in the DDR. Most PP1 regulatory subunits interact with the phosphatase through a conserved binding region termed the RVxF motif, a feature that has facilitated the identification of multiple regulators of the phosphatase in the last few years [35]. Mirroring the great number of functions attributed to PP1, more than 200 of PP1’s interacting proteins (PIPs) involved in a vast number of cellular functions have been described, including glucose metabolism, transcription, cytoskeleton organization, cell cycle and meiosis [36]. PIPs tend to be factors with disordered regions, a feature that allows them to bundle around PP1’s catalytic subunit to form low affinity contacts [33]. One of the most studied and well-defined functions of PP1 is its capacity to revert the spindle assemble checkpoint (SAC) signaling when the spindle microtubules are correctly attached to kinetochores, a function that is attained by its ability to oppose Ipl1/Aurora protein kinase [37]. Interestingly, the fact that cells compromised in both SAC and PP1 activity are still competent to arrest in G2/M suggests that PP1 might be controlling other checkpoints that have been implicated in the transition to mitosis [38]. Today, we know that in addition to the SAC, PP1 is responsible for controlling the meiotic, the morphogenetic, and the DNA damage checkpoint programs [36,38].

One of the first observations that pointed to the PP1 phosphatase as a main regulator of the DDR came from experiments in *Xenopus* eggs extracts. These experiments demonstrated that PP1 and its regulatory element Repo-Man [39,40] are vital to control the threshold of DDR activity in response to a DNA lesion. By looking for a chromatin-bound PP1 complex responsible for suppressing DDR activation, Repo-Man/PP1γ was identified as a holoenzyme with the ability to silence ATM phosphorylation and activation [41] (Figure 1). Supporting this observation, Repo-Man recruits PP1γ to chromatin in unperturbed conditions to dephosphorylate ATM at Ser1981, thus reducing its kinase activity. These experiments clearly demonstrate that Repo-Man/PP1γ controls the threshold of checkpoint activation by opposing ATM. Interestingly, when a DNA lesion is infringed, Repo-Man dissociates from ATM, allowing Ser1981 autophosphorylation and, consequently, the activation of the kinase [42]. Importantly, the levels of Repo-Man expression are frequently elevated in primary tumor tissues and cell lines, rendering cells unresponsive to DNA damage [42]. Moreover, elimination of Repo-Man in late-stage cancer cells reestablishes DDR proficiency in response to genotoxic stress [42]. These results confirm the negative role that PP1 exerts over the damage response and points to the phosphatase as a potential therapeutic target for developing new therapies in cancer treatment.

In addition to Repo-Man, phosphatase 1 nuclear targeting subunit (PNUTS, also known as PP1R10, p99, R111 or CAT 53) is another important PP1-binding factor with implications in the initiation of the damage response. PNUTS can be attached to the α, β and γ isoforms of mammalian PP1 via its RVxF motif [35], targeting PP1 to the nucleus [43]. Once in the nucleus, PNUTS is mainly chromatin-bound and displays multiple pleiotropic functions in transcriptional regulation [44], the control of synaptic signal transduction [45], mitosis exit, and chromosome decondensation [46]. The first implication of PNUTS in the DNA damage checkpoint regulation was demonstrated when it was observed that in the absence of DNA damage, PNUTS-depleted cells experience a prolonged mitotic prophase due to a persistent G2/M arrest. Supporting this observation, PNUTS is rapidly and transiently recruited to DNA damage sites induced by IR. Moreover, in PNUTS-depleted cells exposed to IR, the presence of DNA damage markers like γ-H2AX, 53BP1, replication protein A complex (RPA) and RAD51 are increased, and CHK1 phosphorylation is prolonged due to the hyperactivation of the DDR [24,47]. These results directly link PNUTS with the down-regulation of the DNA damage response through the modulation of the initial stages of the repair pathway, thus buffering checkpoint activity in accordance to the levels of DNA damage infringed to the cell.

### 2.2. Protein Phosphatase 2A

PP2A is a serine/threonine phosphatase that belongs to the PPP family of phosphatases. Like PP1, PP2A is composed of a catalytic subunit, a scaffold subunit, and multiple regulatory elements. The catalytic subunit (PP2Ac) binds to a structural subunit (PP2Aa/PR65) to generate the core of the enzyme. This structure can interact with a wide range of regulatory elements (B55, B56, B72, B130, B48, B93 and B110) to form the active heterotrimeric PP2A holoenzyme complex [48,49]. The large number of identified regulatory elements account for the multiple and diverse PP2A conformations observed, explaining the large number of cellular functions attributed to PP2A [50,51]. In this regard, it has been hypothesized that PP2A might contain more than 80 distinct isoforms in human cells, where each complex could attain a specific localization or target recognition within the cell. Regarding its involvement in the response to a DNA lesion, it has been postulated that PP2A collaborates in the DDR through its interaction with B subunits of the B56 family, such as the B56γ3, B56γ2 and B56δ isoforms. Indeed, it is believed that the formation of these specific PP2A complexes is essential for the role of PP2A in the maintenance of genome integrity in response to DNA damage [52,53].

One of the first realizations of the role of PP2A in the regulation of the DDR came from the observation that its scaffold subunit PP2A-A interacts in vivo with ATM in mammalian cells [54]. The PP2A–ATM interaction is constitutive and takes place in undamaged conditions, suggesting that PP2A restrains ATM activity to avoid unnecessary checkpoint activation in the absence of DNA damage (Figure 1). This model perfectly fits with the observation that undamaged cells treated with the PP2A phosphatase inhibitor okadaic acid [55] induces the autophosphorylation of ATM on Ser1981 [54]. Moreover, the expression of the PP2Ac L199P allele, which acts as a dominant negative by titrating away endogenous regulatory subunits, induces Ser1981 ATM phosphorylation. These experiments demonstrate that PP2A is indeed required for ATM attenuation in the absence of DNA damage [54]. Importantly, after exposure to IR, the interaction between PP2A and ATM becomes weaker, and both complexes dissociates. PP2A–ATM dissociation allows for the autophosphorylation of ATM on Ser1981 and, hence, its own activation, thus triggering the phosphorylation of downstream targets of the DDR [56,57].

Little is known about the molecular mechanism that controls PP2A–ATM interaction in the DDR. Interestingly, it has been proposed that ATM activation in response to replicative stress relies on the suppression of *PPP2R3A*/*B130* expression, a regulatory subunit of the PP2A complex, by the histone deacetylases HDAC1/HDAC2 [58]. This result implies that the negative role that PP2A exert over ATM might be attenuated during the initial steps of the repair process to enhance proficient checkpoint activation. Supporting this view, human ATM directly phosphorylates and specifically regulates B56γ3, B56γ2 and B56δ in response to IR, negatively regulating PP2A activity and directing the complex toward the activation of p53 [59] (Figure 1). Similarly, B56δ is also subjected to CHK1-dependent phosphorylation in *Xenopus* in response to DNA damage [60], suggesting that different kinases might be involved in the negative regulation of PP2A during the generation of a DNA lesion. Though it looks clear the interrelationship between PP2A and ATM in response to DNA damage, the molecular details about this interaction is a question that remains to be addressed. Importantly, PP2A does not exclusively regulate ATM but also ATR. In this regard, it has been demonstrated that the treatment of MCF-7 breast cancer cells with okadaic acid or the use of specific siRNA of PP2A attenuates IR-induced G2/M arrest, and cells display a low level of ATR activation [61].

In addition to the upstream members of the DNA damage checkpoint, PP2A has also been involved in the direct dephosphorylation of downstream targets. One of these examples is CHK2, whose dephosphorylation by PP2A has been demonstrated by different lines of evidence [62,63,64] (Figure 1). PP2A and CHK2 interact in non-DNA damage conditions, an interaction that is lost in response to genotoxic stress. This observation has led to the idea that in the absence of damage, PP2A activity attenuates the DNA damage checkpoint by acting over CHK2 [63]. Another well-known target of PP2A is p53 (Figure 1). It has been demonstrated that PP2A dephosphorylates p53 at Ser37 in response to IR in order to regulate the transcriptional control of the protein [65]. Moreover, the B regulatory subunit B56γ of PP2A is directly involved in p53 dephosphorylation at Thr55. The ablation of B56γ by RNAi results in a hyper-phosphorylation state of p53 at Thr55, a feature that destabilizes the protein and induces *BAX* expression and cell apoptosis [52]. Finally, the activity of PP2A is not only required for controlling the levels of DNA damage checkpoint activation in response to genotoxic stress but also in the maintenance of the signal during the repair process. Accordingly, PP2A-B55α interacts and dephosphorylates PLK1 at Thr210 in an ATM/ATR-dependent manner in response to DNA damage [66,67]. Since PLK1 is required to promote the G2/M transition, the PP2A-dependent inhibition of PLK1 ensures a robust G2 blockage in response to genotoxic stress (Figure 1).

### 2.3. Protein Phosphatase 4

Mammalian PP4 was originally discovered in the nineties as a ubiquitous serine/threonine phosphatase that regulates many cellular functions and that contains a great similarity with its family members PP1 and PP2A [68,69]. In fact, for a long time, it was thought that several PP4 subunits were part of the PP2A holoenzyme. Today, we know that PP4 is an independent phosphatase complex that plays critical and specific roles independently from both PP1 and PP2A. Like other phosphatases, PP4 is comprised by a catalytic subunit and several regulatory elements that provide the holoenzyme with the ability to recognize specific targets. Little is known about PP4’s molecular structure; however, the amino-acid sequence of its main subunit in higher eukaryotes is 41% and 65% identical to that of PP1 and PP2A, respectively, suggesting that the structural core conformation of these phosphatases might be similar. In mammalian cells, the main catalytic subunit PPP4 is accompanied by two structurally distinct regulatory subunits, called R1 and R2. Similarly to human cells, the budding yeast PP4 holoenzyme is composed of the catalytic subunit Pph3 and two regulatory elements termed Psy2 and Psy4. In budding yeast, Pph3 shares a great similarity in amino-acid sequence with both catalytic subunits of PP2A (Pph21 and Pph22), indicating that, in principle, it may form part of the PP2A holoenzyme [68,69]. In terms of function, PP4 has been linked to multiple cellular events, such as organelle and spliceosomal assembly, cell signaling and growth, chromatin remodeling and centromere pairing in meiosis [70].

Little is known about the role of PP4 in the initial events of the DDR. In *Saccharomyces cerevisiae,* PP4 has been involved in the activation of the DNA damage checkpoint in response to replication stress. In two independent screenings to identify suppressors of the *mec1-100* lethality (a Mec1 allele that displays a delayed activation of Rad53 in the S phase [71]) on hydroxyurea (HU), it was found that both *pph3Δ* and *psy2Δ* cells restore Mec1 deficiency, and it was demonstrated that most of the *mec1-100*-compromised targets were PP4-regulated [72]. Moreover, PP4 dephosphorylates Mec1 itself at Ser1991, regulating its activity and conferring damage sensitivity, an event that takes place due to the physical interaction between both complexes at the sites of replication fork collapse and DSBs [72] (Figure 1). In all, it is tempting to speculate that the interconnection between Mec1 and PP4 in response to genotoxic stress ensures the correct balance in protein phosphorylation of multiple DDR components at any given time of the damage response. How the equilibrium between the activity of Mec1-Ddc2 and PP4 is achieved within the complex along the different stages of the DDR is a fascinating question for the future.

### 2.4. Protein Phosphatase 5

PP5 is another member of the PPP family of serine/threonine phosphatases. Since its discovery, PP5 has been related to a wide range of cellular processes, including growth, differentiation, cell cycle control, the regulation of ion channels, heat shock response, and steroid receptor signaling [73]. Interestingly, the sequence homology between PP5 and PP1, PP2A or PP4 is rather low, suggesting that this phosphatase might be independently acting over distinctive phospho-substrates from other members of the family. In addition, unlike other PPP family members, which form holoenzymes composed of a great number of regulatory elements, PP5 is a single subunit enzyme that uses its N-terminal TRP domain to achieve substrate recognition and activity regulation [74,75,76]. At the structural level, PP5 contains several conserved elements of the PPP group of phosphatases that provide substrate recognition and interaction. However, despite this highly conserved structure, PP5 encloses specific elements for substrate recognition at the catalytic domain [77]. Together with PP1, PP2A and PP4, PP5 has also been shown to cooperate in the accurate regulation of the DNA damage checkpoint. In mammalian HEK 293T and HeLa cells, both ATM and ATR interact with PP5 in response to DNA damage, an interaction that endorses the full phosphorylation and activation of the kinases [78,79,80] (Figure 1). Corroborating these results, PP5-deficient mouse embryonic fibroblasts (MEFs) display a significant defect in DNA damage checkpoint activation in response to IR due to the alteration of the ATM-mediated signaling [81]. How is the PP5-dependent activation of ATM/ATR executed? New evidences have demonstrated that ATM dimerization can inhibits its kinase activity and that the early autophosphorylation of Ser1981 in response to DNA damage induces their dissociation into active monomeric forms [57]. This observation has pointed to the control of Ser1981 phosphorylation as the main mechanism that regulates ATM activity in response to a DNA lesion. Interestingly, Ser1981 phosphorylation is diminished in cells that lack PP5 activity, suggesting that this phosphatase is indeed an important regulator of ATM activation during the initial steps of the DDR by modulating the kinase dimerization [78]. It is important to remark that PP5 can also regulate the DNA damage response by directly dephosphorylating ATM-downstream targets. In this regard, it has been reported that PP5 directly binds with and dephosphorylates p53 at multiple serine/threonine residues in mice, thus inhibiting p53-mediated transcriptional activity [82] (Figure 1). Taking into account that p53 strongly represses PP5 transcription, it appears that the reciprocal interplay between PP5 and p53 might provide a feedback mechanism for the accurate activation of the DNA damage response [82]. Similarly to ATM, PP5 plays as well a critical role in ATR-mediated repair of UV-induced DNA damage [80]. Ratifying these data, PP5-deficient MEFs have an increased sensitivity to UV light, hydroxyurea, and camptothecin, as well as prolonged CHK1 phosphorylation and an increased phosphorylation and protein levels of p53 [83].

Despite the clear evidence pointing out PP5 as a positive regulator of the ATM–ATR pathways, it has recently been found that the disruption of ATM Ser1981 in HeLa cells has no effect on its catalytic activity and that PP5 does not enhance the phosphorylation of this residue after DNA damage [84]. These results indicate that PP5 might be controlling the steady state activity of ATM by acting over other phospho-residues or substrates. Accordingly, in a two-hybrid screening designed to isolate PP5 interacting proteins, it was established that PP5 interacts with a cluster of six potential phosphorylation sites of DNA-protein kinase (PK), including Thr2609 [84]. Interestingly, while the overexpression of PP5 after IR exposure produces a PP5-dependent dephosphorylation of DNA–PK at Thr2609, no effect has been observed in ATM Ser1981 under the same conditions [84]. These results suggest that PP5 enhances DDR activation in response to DNA damage, mainly by modulating the DNA–PK pathway. 

### 2.5. CDC14

Another phosphatase that has been involved in the timely activation of the DNA damage response is the cell division cycle 14 (CDC14). CDC14 is one of the most studied families within the DUSPs, proteins that are characterized by their ability to dephosphorylate both phosphotyrosine and phosphoserine/phosphothreonine residues in their substrates [85]. A specific characteristic of CDC14 is its predisposition to dephosphorylate targets that have previously been phosphorylated by the CDK. Up to date, multiple roles of CDC14 have been described in cytokinesis [86,87], chromosome segregation [88,89], transcription [88,90,91], centrosome duplication [89], ciliogenesis [19,20,21,92], DNA repair [23], and in resolving linked DNA intermediates [19,20,21].

During the last few years, it has become evident that the CDK is a master regulator of the DDR. Therefore, it is intuitive to think that CDC14 might also have relevant roles in the damage response by counteracting CDK substrates. The first realization of a function of CDC14 in the regulation of the DNA damage checkpoint activation came from the observation that Flp1 (fission yeast ortholog of CDC14) translocates from the nucleolus to the nucleoplasm in response to replication stress induced by HU treatment [93]. Nowadays, it is well known that the nucleolar release of Flp1 is mediated by the checkpoint kinase Cds1 (homologue of Rad53 and CHK2 in *S. cerevisiae* and humans, respectively), which colocalized to stalled replication forks after replication stress. Interestingly, the Flp1 nucleolar release induces a positive feedback loop that ends up with the complete activation of Cds1, thus triggering a proficient activation of the DNA damage checkpoint [93] (Figure 1). It is important to note that CDC14 liberation from the nucleolus in response to DNA damage is not restricted to *Schizosaccharomyces pombe.* It has been demonstrated that mammalian CDC14B is also excluded from the nucleolus in response to DNA damage [94]. Liberated CDC14B enhances PLK1 degradation by the ubiquitin ligase APC/C-CDH1, thus stabilizing the DNA damage activator Claspin and the cell cycle inhibitor WEE1, resulting in the activation of the G2/M arrest [94] (Figure 1).

Intriguing, the knockdown of *CDC14A* or *CDC14B* in chicken DT40, human HCT116, and human telomerase reverse transcription-immortalized retinal pigment epithelial cells are proficient for the activation of the damage checkpoint in response to irradiation, but they exhibit DNA repair defects [95]. The same results are recapitulated in CDC14-deficient MEFs and in the budding yeast exposed to DNA damage [23,96] (see Section 3.4). These contradictory results suggest that the role of CDC14 in the DDR might differ depending on the organism and/or cell type analyzed, complicating the study of the phosphatase in response to a DNA lesion.

## 3. Role of Protein Phosphatases in the Repair of a DNA Lesion

Once a DNA lesion has been detected, cells must coordinate the activation of the DNA damage checkpoint with the repair of the DNA adduct. DSBs are repaired either by the direct ligation of the broken ends (NHEJ) or by repair pathways that rely on the searching of an intact homologue sequence that serves as donor for the repair (HR) [97] (Figure 2). Homology-dependent DSB repair is always initiated by the nucleolytic degradation of the 5′-end strand of the DNA molecule to produce a single-stranded DNA tail, a process generically known as resection [98,99]. Resection is executed in a two-step process: in the first step, the MRX/MRN complexes (Mre11–Rad50–Xrs2 in yeast/MRE11–RAD50–NBS1 in mammals) together with Sae2/CtIP generate a short 3′ overhang ssDNA (single-stranded DNA); in a second step, two redundant pathways involving the Sgs1–Top3–Rmi1–Dna2/BLM–TOPOIIIα–RMI1–DNA2 complex and the nuclease Exo1/EXO1 extend resection to give rise to long ssDNA tracks (Figure 2) [100]. These ssDNA tracks are quickly covered by RPA to protect the DNA filament from other nucleases that might degrade it. This protein-ssDNA structure, termed the nucleofilament, has the ability to search and invade into a homologous DNA sequence that serves as donor for DNA polymerases to copy the lost information and regenerate the broken DNA [101].

Lately, it has been well documented that multiple factors involved in DNA repair are activated by phosphorylation events [10,102]. Therefore, it is intuitive to think that protein phosphatases might have a role in restoring the balance imposed by the DDR-kinases in the damage response. Indeed, it has been demonstrated that the orchestration of the kinase/phosphatase activity along the different stages encompassed in the repair cycle is essential to ensure the accurate restoration of the DNA lesion. Accordingly, several pieces of evidence have pointed out protein phosphatases as keystone members of the repair process, confirming their vital contribution in the regeneration of a DNA adduct. In this section, we evaluate the molecular determinants of protein phosphorylation/dephosphorylation of different DNA repair substrates, the exact composition of each phosphatase holoenzyme responsible for the accurate execution of the repair process, and the biological significance of their functions in the physical restoration of a DNA lesion.

### 3.1. Protein Phosphatase 1

In addition to the role of PP1 in controlling the DNA damage checkpoint response, this phosphatase also participates in DNA repair by modulating several factors of the repair machinery. The breast and ovarian cancer type 1 susceptibility protein BRCA1 is involved in the execution and coordination of various aspects of the DNA damage response, including DNA repair and checkpoint control [103,104,105]. Regarding its function in DNA repair, BRCA1 binds and stimulates phosphorylated CtIP [106] and MRN [107], thus enhancing the formation of the ssDNA necessary for the execution of recombination repair pathways (Figure 2). Interestingly, both PP1α and BRCA1 physically interact in vitro and in vivo through the RVxF binding motif of PP1α [108]. Moreover, PP1α specifically dephosphorylates hCDS1/CHK2-phosphorylated BRCA1, even though this regulation might be reciprocal because BRCA1 can also inhibit PP1α [108]. The physiologically significance of the PP1α–BRCA1 interaction has been under debate for the last few years. Though it seems clear that the PP1α interaction with BRCA1 is required for survival-enhancement and HR execution following IR [108,109], the molecular basis for this function is not yet well understood. However, the fact that the formation of a stable PP1α–BRCA1 complex is required for the correct distribution of BRCA1 and RAD51 to damage sites [109], suggesting that PP1α might direct BRCA1 to the DNA lesion (Figure 2). To date, it is unclear how this mechanism takes place, but the fact that a PP1α non-binding mutant of BRCA1 displays a marked increase in its phosphorylation levels suggests that BRCA1 is a target for dephosphorylation within the PP1α–BRCA1 holoenzyme complex [110]. Interestingly, the role of PP1 in DNA repair is not restricted to homology-dependent DNA repair but also to NHEJ. In this regard, it has been shown in both *Xenopus* and human cells that PP1 interacts, dephosphorylates, and activates DNA–PKcs in response to DNA damage [111] (Figure 2).

It is important to remark that the binding of different PP1 regulators along the damage response is crucial for the timely orchestration of the DNA repair events. In this matter, it has been postulated that the covalent fusion of PP1 with the regulatory subunit NIPP1 results in the formation of RNA–DNA hybrids (R-loops), enhanced chromatin compaction, slow replication fork progression and inefficient DNA repair [112]. Another well-known PP1 regulator factor that has been involved in DNA repair is PNUTS. PNUTS binds with DNA–PK and enhances the phosphorylation of DNA–PKcs at Ser2056 and Thr2609, positively regulating NHEJ activity in response to DNA damage [111]. These results entail that in response to DNA damage, PP1 is subjected to a fine-tune regulation by its regulatory subunits in order to mediate the precise execution of different repair pathways.

### 3.2. Protein Phosphatase 2A

The first realization of a role of PP2A in DNA repair came from the observation that HeLa cell extracts treated with okadaic acid resulted in a drastic deficiency in NER execution [13]. Soon after, it was found that PP2A was targeting several components of the DNA–PK complex in in vitro experiments [113], indicating that the role of PP2A in DNA repair can also been extended to NHEJ (Figure 2). The phosphorylation of DNA–PKcs, Ku70 and Ku80 has been correlated with the inactivation of the kinase activity of the complex. Interestingly, the PP2A dephosphorylation of DNA–PK restores its kinase activity in vitro. Moreover, the treatment of human lymphoblastoid cells with PP2A inhibitors (as okadaic acid or fostriecin) significantly decreases DNA–PK activity, demonstrating the ability of PP2A to activate DNA–PK in vivo [113]. Supporting these results, the elimination of twins (B regulatory subunit B/B55 of PP2A in *Drosophila melanoganster*) induces the accumulation of persistent γH2AX foci, high levels of Ku70 phosphorylation, and DNA repair failures that lead to the appearance of chromosome aberrations [114]. These results validate that PP2A forms part of an essential mechanism that positively regulates DNA repair by the NHEJ pathway and suggests that its mechanism of action is evolutionarily conserved among eukaryotes.

In addition to the role of PP2A in regulating NEHJ, the phosphatase has also been implicated in HR. In this regard, it has been shown that PP2A coimmunoprecipitates and colocalizes at DNA damage foci with γ-H2AX [115]. Taking into account that elimination of PP2A activity affects both the dissolution of γ-H2AX foci and the efficiency of DNA repair, it is reasonable to think that the phosphatase cooperates in recombinational DNA repair by stimulating γ-H2AX dephosphorylation (Figure 2). Importantly, this effect is independent of the role of PP2A in regulating ATM/ATR or DNA–PK [115], indicating that PP2A may have multiple functions in the response to DNA damage. Similar results have been found when disrupting the B55α or B56 regulatory subunits, demonstrating that the formation of a PP2A holoenzyme is a prerequisite for the accurate function of the phosphatase in recombinational repair pathways [116].

Another target of PP2A with a role in the repair of a DSB is RPA (Figure 2). It is known that in response to DNA damage, ATM and ATR phosphorylate RPA at positions Thr21 and Ser33. RPA phosphorylation is required to inhibit DNA replication and to recruit other components of the repair machinery to the DNA lesion. In cells recovering from HU treatment, PP2A is essential to down-regulate RPA phosphorylation. A lack of the PP2A-dependent dephosphorylation of RPA causes an increased sensitivity to HU. Importantly, a phosphomimetic version of RPA precisely activates/deactivates the DNA damage checkpoint but is affected in DNA repair [117]. These results indicate that the role of RPA dephosphorylation by PP2A is exclusively restricted to the accurate restoration of the DNA lesion.

Finally, it is important to remark that PP2A might be indirectly involved in the restoration of a DNA break through its ability to negatively regulate the DNA damage checkpoint (see Section 2.2). It has been demonstrated that cells lacking the heterotrimeric PPP2R2A complex develop high levels of ATM and CHK2 phosphorylation, resulting in a G1 to S phase arrest and the down-regulation of canonical HR factors such as BRCA1 and RAD51 [118]. In these circumstances, cells are affected in the fidelity of the HR pathway and are sensitized to PARP inhibitors [118]. These data suggest that the tight regulation of DDR targets by PP2A in response to DNA damage is essential for the timely execution of the pathway. Whether this PP2A role is directly related to its capacity to modulate checkpoint components or if is an indirect consequence of its cell cycle regulation function is an open question.

### 3.3. Protein Phosphatase 4

Similarly to PP2A, PP4 is also involved in DNA repair by acting over multiple factors that are implicated in the restoration of a DNA lesion. Curiously, most of the PP4 targets identified in response to a DNA lesion are also PP2A substrates, suggesting that both phosphatases might have redundant functions in DNA repair. The first realization of the role of PP4 in the repair of DNA lesions came from the observation that Psy2, the Pph3 regulatory subunit of the *S. cerevisiae* PP4 complex, was required for Rad53 dephosphorylation during treatment and recovery from methyl methanesulfonate (MMS). In the absence of Psy2, both the dephosphorylation of Rad53 and resumption from replicative stress are delayed, even though genome replication is eventually completed by the firing of late origins of replication. These results indicate that the PP4 complex is directly implicated in the stabilization of stalled or collapsed replication forks by acting over Rad53 [119]. Recently, it has been demonstrated that PP4-dependent Rad53 dephosphorylation is required not only for replication stress conditions but also during the repair of a DSB by HR. The role of Pph3 in recombinational DNA repair entails Rad53 dephosphorylation during the initial stages of the DDR, an event that mitigates the negative effect that Rad9 exerts over the Sgs1/Dna2 exonuclease complex, thus boosting resection [22] (Figure 2). In addition to Rad53, Pph3 is also required to dephosphorylate γ-H2A in vivo and in vitro in *S. cerevisiae* [120]. However, while PP4-dependent γ-H2A dephosphorylation has been linked to checkpoint deactivation and cell cycle re-entry (see Section 4.3), there is no clear evidence that indicates that this function could be needed for DNA repair in budding yeast. Finally, it is important to mention that PP4-dependent roles in DNA repair might be redundant with PP2C since the depletion of both Pph3 and Ptc2/3 in the budding yeast exhibits a synergistic sensitivity to camptothecin, HU and MMS, and it endorses a significant decrease in the restoration of an HO-induced DSB [121].

As in *S. cerevisiae,* human U2OS cells lacking PP4C activity develop defects in the steady state phosphorylation of γ-H2AX concomitantly with a persistent activation of the DNA damage checkpoint [122]. Supporting these observations, the depletion of *PP4C*, *PP4R2* and *PP4R3β* in HeLa cells also develops high levels of γ-H2AX phosphorylation [123] (Figure 2). Importantly, the inhibition of PP4 activity results in an inefficient repair of DNA replication-mediated lesions [123], directly linking the PP4-dependent dephosphorylation of γ-H2AX with DNA repair in human cells. Mirroring the high level of PP4–PP2A redundancy in the DDR, human PP4 is also capable of dephosphorylating RPA. PP4 efficiently dephosphorylates RPA in vitro, and its silencing in vivo alters the kinetics and pattern of RPA phosphorylation. Importantly, the depletion of PP4R2 activity affects the execution of the HR pathway due to the inefficient loading of RAD51 to the breaks [124]. Taking into account that RPA and RAD51 compete for ssDNA binding in both *S. cerevisiae* and human cells [125,126,127], it is tempting to speculate that RPA dephosphorylation by PP4 might control recombinational DNA repair by modulating the recruitment of recombinational factors to ssDNA. 

### 3.4. CDC14

The role of CDC14 in the DDR has been at the center of attention and debate for the last few years. Even though the implication of this phosphatase in the damage response was originally associated to its ability to modulate checkpoint activation, recent results have suggested that CDC14 mostly operates at the DNA repair level. The first realization of a role of CDC14 in DSB repair came from studies that used knockouts of *CDC14A* and *CDC14B* in chicken DT40 cells, human HCT116 somatic cells, and human telomerase reverse transcription-immortalized retinal pigment epithelial cells. While disruption of *CDC14A* and *CDC14B* in these biological systems does not affect the execution of the G2/M checkpoint arrest, these cells develop a profound defect in the repair of DNA lesions. Accordingly, the exposure of cells lacking *CDC14A* or *CDC14B* to IR increases the number of γ-H2AX foci, and the DNA breaks persists for longer, suggesting a direct role of the phosphatases at the repair level [95]. However, these cells completely support a stable DNA damage checkpoint activation and trigger a proficient G2/M arrest. Supporting these results, *CDC14B*-deficient mouse embryonic fibroblasts exposed to genotoxic stress also develop high levels of endogenous DNA damage and trigger senescence without any interference with the DNA damage checkpoint machinery [96]. It is important to remark that MEFs cells lacking CDC14B present defects in repairing IR-induced DSBs only at late passages, when CDC14A levels are low [128], suggesting that both phosphatase isoforms might collaborate in DNA repair by sharing similar functions.

Importantly, it seems that the role of CDC14 in the repair of DNA lesions is an evolutionarily conserved feature in eukaryotes. CDC14 is released from the nucleolus and activated in response to several sources of DNA damage in *S. cerevisiae, S. pombe* and humans [23,89,93,129]. In budding yeast, the nucleolar release of Cdc14 is required for the stabilization of the mitotic spindle by acting over the spindle pole body (SPB) component Spc110, a fundamental event that enhances the recruitment of DNA lesions to the vicinity of the SPBs for their repair by HR [23]. Though it seems evident the direct involvement of the phosphatase in DNA repair, Cdc14 has also been implicated in the resolution of recombinant intermediates that are generated during the process by acting over the resolvase Yen1 [19,20,21] (Figure 2). The CDK-dependent phosphorylation of Yen1 during the S phase promotes its nuclear exclusion and inhibits its catalytic activity by reducing its DNA binding capacity. In anaphase, the nucleolar release of Cdc14 counteracts CDK-dependent Yen1 phosphorylation, thus enhancing its transport into the nucleoplasm [19,20,21]. The timely anaphase activation of Yen1 by Cdc14 ensures the elimination of any recombination intermediate that is left unresolved during the repair of a DNA lesion, ensuring the accurate distribution of the genetic material in mitosis.

Overall, these results indicate that, at least in the budding yeast, Cdc14 collaborates in the restoration of a DNA lesion at two different levels: the repair of a DNA lesion and the resolution of recombination intermediates. How does Cdc14 regulate these two distinctive and temporally separated events of the repair process? It has been demonstrated that the DNA damage-dependent nucleolar release of Cdc14 is a transient event that is restricted to the nucleoplasm and to the G2/M checkpoint arrest [23]. This suggests that the nucleoplasmic redistribution of Cdc14 in response to DNA damage might account for the specific regulation of the DNA repair machinery. Accordingly, the induction of an HO-induced DSB stimulates the formation of discrete Cdc14 foci that colocalize with Ddc2 [23], suggesting a physical interaction of the phosphatase with components of the DNA repair pathway. On the other hand, the DNA damage-independent cytoplasmic release of Cdc14 during the metaphase-to-anaphase transition might account for the activation and translocation of Yen1 to the nucleus, thus ensuring the correct resolution of the DNA entanglements that are left unresolved during the repair process. Thus, the distinctive Cdc14 redistribution observed during the successive steps of the DNA repair pathway might account for its substrate specificity. It is important to remark that, due to the functional redundancy of CDC14 between different model organisms, it is feasible that their functions in DNA repair might be conserved between different species. 

## 4. Protein Phosphatases in DNA Damage Checkpoint Deactivation and Cell Cycle Recovery

Once the DNA lesion has been repaired, cells have to deactivate the DNA damage checkpoint to allow for cell cycle resumption. This process, generically termed as DNA damage checkpoint recovery, relies mainly on the Polo-like kinase 1 protein PLK1. In human cells, the PLK1-dependent phosphorylation of both WEE1 and Claspin targets them for proteasomal degradation, thus restraining the negative effect that these factors exert over the CDK activity [130,131] (Figure 3). A similar process takes place in *S. cerevisiae*, since Rad53 dephosphorylation and attenuation by Cdc5 (budding yeast homologue of human PLK1) allows cells to re-enter in the cell cycle in response to an irreparable DNA break. Even though the role of Cdc5/PLK1 is vital to endorse a full checkpoint recovery, its individual activity is not enough to override the DNA damage checkpoint. Since most of the DNA damage checkpoint routes rely on the active phosphorylation of their components, it is intuitive to think that dephosphorylation events by DNA-damage specific phosphatases might participate in the silencing of the damage response when the DNA lesion has been fixed. Indeed, recent discoveries have put forward protein phosphatases as active enzymes that are essential to re-establish the phosphorylation status of multiple DDR targets during cell recovery. Therefore, in this section, we focus our attention on the role of different phosphatases in DNA damage checkpoint recovery, their targets in the process, and the fundamental principles behind their regulation during cell cycle re-entry.

### 4.1. Protein Phosphatase 1

The role of PP1 in cell cycle recovery upon exposure to DNA damage was initially anticipated in a *S. pombe* screening to identify genes that, when overexpressed, were able to override the DNA damage checkpoint and re-enter into the cell cycle even in the presence of DNA damage. In this screening, the overexpression of Dis2 (main subunit of the PP1 complex in the fission yeast) was enough to abrogate Chk1 phosphorylation and activation in vivo [132] (Figure 3). Moreover, the inactivation of Dis2 renders cells to a prolonged G2/M arrest in response to MMS or the UV-mimic drug 4-NQO (4-nitroquinoline-1-oxide) due to their incapacity to properly dephosphorylate Chk1 upon repair [132]. These data indicate that in fission yeast, PP1 is responsible for Chk1 dephosphorylation following DNA repair, a feature that enhances DNA damage checkpoint silencing and, thus, cell cycle re-entry. Interestingly, while fission yeast PP1 is not required for checkpoint recovery upon exposure to replication stress induced by HU treatment, the disruption of Glc7 in the budding yeast (main catalytic subunit of the PP1 complex) restrains Rad53 dephosphorylation and recovery from replication fork stalling in response to HU [133]. These results imply that different organisms use different phosphatases to counterbalance the effect of DDR-kinases during checkpoint recovery and that this activity depends on the type of DNA damage infringed. Additionally, the role of PP1 in checkpoint deactivation might be evolutionarily conserved, since the disruption of Sds22 (regulatory subunit of the PP1 complex in *Candida albicans*) endorses a hyper-phosphorylation state of Rad53 upon MMS treatment [134].

In humans, PP1 is also involved in checkpoint recovery upon genotoxic stress by dephosphorylating p53 at Ser15 and Ser37 and by attenuating its transcriptional activity [135] (Figure 3). Since p53 has been directly involved in the expression of the CDK inhibitor p21, it is tempting to speculate that PP1-dependent p53 inactivation might have a direct role in checkpoint inactivation by enhancing CDK activity. Supporting this notion, it has been reported that several PP1 regulators such as PNUTS, p53BP2 or GADD34 have the ability to modulate the steady state phosphorylation of p53 [136,137,138]. Still, the influence of PP1 in reactivating the CDK during checkpoint recovery is not only restricted to p53 regulation but also to CDC25 (Figure 3). As mentioned above, CDC25 stimulates mitotic entry by eliminating CDK inhibitory phosphorylation [139,140]. In response to DNA damage, both CHK1 and CHK2 phosphorylate CDC25C to exclude the protein from the nucleus, thus restraining CDK activity and consequently stimulating the G2/M checkpoint arrest. In *Xenopus*, it has been documented that PP1 efficiently dephosphorylates CDC25 at Ser287, a feature that stimulates its phosphatase activity and subsequently mitotic entry [139]. This has led to the tantalizing hypothesis that PP1 might also collaborate in DNA damage checkpoint recovery by enhancing CDK activity through the direct activation of the CDC25 phosphatase.

Finally, in *S. cerevisiae,* PP1 dephosphorylates γ-H2A in vitro and is required to counteract its phosphorylation in vivo, an essential function required for checkpoint recovery in response to replication fork stalling [133] (Figure 3). Moreover, the PP1-dependent dephosphorylation of γ-H2A is strictly required for cell cycle re-entry since cells lacking Glc7 activity are.competent to endorse a proficient DNA damage checkpoint activation and a correct execution of recombinational DNA repair mechanisms [133].

### 4.2. Protein Phosphatase 2A

The role of PP2A in an undamaged cell cycle progression was originally discovered when realizing that its lack of activity resulted in a precocious mitotic entry of fission yeast cells [141]. The same conclusion was obtained in *S. cerevisiae*, in which Pph21 and Pph22 (PP2A catalytic subunits) and Cdc55 (PP2A regulatory element) were isolated in a screening to identify genes that were toxic to *cdc5-1* mutants when overexpressed [142]. Though the ability of PP2A to restrain mitotic entry has been conserved in evolution, the molecular mechanisms that exert this function differ between different organisms. While in *Xenopus,* PP2A regulates mitotic entry by acting over CDC25 [143], in *S. cerevisiae,* the final acceptor of PP2A seems to be the WEE1 homologue Swe1 [144]. By contrast, in *S. pombe* both Wee1 and Cdc25 are controlled by PP2A and its regulatory subunit Pab1 [145].

Recently, it has become clear that PP2A activity is negatively regulated by the Greatwall kinase [146]. The mitotic activation of Greatwall induces a negative feedback loop that ends up with PP2A inhibition, allowing cells to enter in mitosis [147] in a pathway that implies the phosphorylation of the PP2A inhibitor ARPP19/ENSA [148,149]. How is this mechanism regulated in response to DNA damage? It has been hypothesized that in response to DNA damage, both CDK and PP2A activity might form part of a balance that regulates progression into mitosis. In the absence of DNA damage, CDK activity eventually overcomes that of PP2A and triggers mitotic entry. On the other hand, the down-regulation of the CDK activity attained by the DDR activation in response to a DNA lesion might drive the balance to PP2A, thus restraining cells from entering into mitosis [150]. In this scenario, it is tempting to propose that Greatwall inactivation in response to DNA damage could relive PP2A inhibition and consequently endorse a proficient G2/M arrest by dephosphorylating CDK targets. When the repair of the DNA lesion has been attained, Greatwall re-activation could promote PP2A inhibition and cell cycle reactivation. This hypothesis is in line with recent discoveries in human cells that have demonstrated that modulation of the PP2A-B55/ENSA/Greatwall pathway is indeed required for cell cycle resumption in response to DNA damage. The interaction between PLK1 and PP2A-B55α or PP2A-B55δ is stimulated during recovery from DNA damage. In addition, the disruption of PP2A activity promotes PLK1 phosphorylation and progression into mitosis [151]. In line with these findings, MASTL, the human homologue of Greatwall, is not required either for DDR activation nor DNA repair, but it is required for controlling the timing of mitosis re-entry and the subsequent fate of the recovering cells [152]. Importantly, the down-regulation of MASTL or ARPP19/ENSA delayed mitotic entry in response to DNA damage, confirming the involvement of the kinase in the regulation of the G2/M arrest. In accordance to this model, studies of *Xenopus* have shown that Greatwall is actively inhibited by the DDR in response to DNA damage, thus allowing for PP2A activation during the initial stages of the repair process [41]. Once the DNA lesion has been repaired, the silencing of the DDR signaling reactivates Greatwall to inhibit PP2A activity, thus enhancing mitosis entry [41] (Figure 3).

### 4.3. Protein Phosphatase 4

The role of the PP4 phosphatase complex in DNA damage recovery has been greatly studied in the last few years. One of the most important targets of the budding yeast PP4 complex during DNA damage recovery is probably Rad53. It has been demonstrated that Pph3 binds to its regulatory element Psy2 to efficiently dephosphorylate activated Rad53 during recuperation from MMS [119] (Figure 3). This interaction is necessary to enhance replication fork restart, since the elimination of Psy2 drastically affects DNA synthesis in response to MMS. However, under these circumstances, while cells ultimately complete genome replication by initiating late origins of replication, the levels of Rad53 phosphorylation remain abnormally high. This result suggests that PP4 activity over Rad53 is not only restricted to timely replicate the genomic material but also to enhance Rad53 dephosphorylation and cell cycle resumption in response to alkylating agents.

Another well-known target of PP4 with implications in DNA damage checkpoint recovery is γ-H2AX (Figure 3). It has been reported that PP4 is also required for cell cycle recovery in response to DNA damage in *Drosophila* [114]. However, the fact that lack of PP4 activity impairs the proper resolution of γ-H2AX foci [114] leaves it unclear whether the cell cycle re-entry phenotype is exclusively due to the role of PP4 in cell cycle reactivation or if it is a consequence of the inefficient repair attained in the absence of the phosphatase. Interestingly, since PP4 in *S. cerevisiae* is not required for DNA repair by HR in an inter-chromosomal DNA repair system and that the dephosphorylation of γ-H2A takes place after its removal from the DNA, it seems that PP4’s role in the DDR is strictly related to its capacity to endorse a proficient DNA damage checkpoint silencing upon repair [120]. Similar results have been found in human cells, even though in this case, PP4 removes γ-H2AX at damaged sites and from undamaged chromatin [122]. Importantly, the hyper-phosphorylation of γ-H2AX in the absence of human PP4 correlates with higher levels of MDC1 (mediator for DNA damage checkpoint 1) bound to γ-H2AX at the sites of damage [122]. Taking into account that MDC1 binds specifically to phosphorylated γ-H2AX [153], it appears that PP4 dephosphorylation of γ-H2AX excludes MDC1 from damage sites, thus strengthening the inactivation of the DNA damage checkpoint. Supporting this hypothesis, a mutation of H2A at serine 129 to alanine fully restores the ability of *S. cerevisiae pph3Δ* cells to turn off checkpoint signaling in a timely manner [120]. However, the dephosphorylation of γ-H2A by PP4 does not account for the phenotype observed in *pph3Δ* cells since a double mutant *pph3Δ hta1-S129A* presents an additive sensitivity to MMS treatment [119]. Moreover, disruption of Psy4 (regulatory subunit of PP4) increases H2A phospho-levels without affecting Rad53 phosphorylation state, mechanistically separating both events [119].

### 4.4. WIP1

*WIP1* was originally identified in human cells as a transcript that was expressed in response to IR in a p53-dependent manner. The biochemical characterization of the protein revealed a strong similarity with type 2C phosphatase, including Mg2+ dependence and insensitivity to okadaic acid [154]. It was soon realized that WIP1 presented an important role in DNA damage checkpoint inactivation upon repair. This is attained by the ability of the phosphatase to dephosphorylate p53 [155,156], CHK1 [156,157] and CHK2 [158,159] (Figure 3), thus down-regulating DNA damage checkpoint signaling and enhancing cell cycle recovery. In mice, WIP1 dephosphorylates ATM at Ser1981, a critical site for ATM monomerization and activation (see Section 2.1, Section 2.2 and Section 2.4), as soon as the damaged DNA has been repaired [160] (Figure 3). WIP1’s capacity to silence the DNA damage checkpoint is not only attained by down-regulating the activity of the main kinases involved in the activation of the response but also through the control of their stability. In this regard, it has been demonstrated in humans that the WIP1-dependent dephosphorylation of the E3 ubiquitin ligase MDM2 stabilizes the protein, thus facilitating p53 ubiquitination and degradation [161] (Figure 3). Interestingly, p53 inactivation by WIP1 is not only required to strengthen the DNA damage checkpoint silencing upon repair but also to retain cellular competence to divide by enhancing a steady-state transcription of mitotic inducers during the G2/M arrest [162]. This infers that WIP1 needs to be active through the damage response to confer a proficient checkpoint recovery after the DNA lesion has been repaired. This idea is in line with previous evidence that demonstrating that p53 is subjected to pulses in its activity during the execution of the DNA damage response that depends on ATM and WIP1 [163]. Another well-known target of WIP1 is γ-H2AX, whose dephosphorylation by the phosphatase stimulates MDC1 displacement from damage foci and prevents the activation of the DNA damage checkpoint [164,165,166] (Figure 3). It is important to remark that the precocious dephosphorylation of γ-H2AX by WIP1 has been related to an inefficient execution of the DNA repair process. This is due to a disruption in the recruitment of important DNA repair factors to DSBs, a feature that restrains the proper accomplishment of the DNA damage repair pathway [165,166].

As other DDR-related phosphatases, the activity of WIP1 is tightly regulated along the damage response in order to avoid precocious dephosphorylation of DNA repair factors that could affect the correct restoration of the DNA lesion. At the transcriptional level, *WIP1* is regulated by the expression of the microRNA miR-16 during the initial events of the DDR. In response to DNA damage, miR-16 is quickly induced to target *WIP1* mRNA, thus negatively regulating the expression levels of *WIP1*. This mechanism ensures the down-regulation of WIP1 immediately after the generation of the DNA lesion, preventing the premature inactivation of the ATM/ATR pathway and allowing for the proper execution of DNA repair events [167]. Importantly, the transcriptional regulation of *WIP1* expression is not the only mechanism that ensures the timely activation of the phosphatase. In undamaged cells, WIP1 is constitutively phosphorylated by the homeodomain-interacting protein kinase 2 (HIPK2) targeting the phosphatase to proteasomal degradation [168]. In response to DNA damage by IR, the ATM-dependent phosphorylation of HIPK2 restrains its interaction with WIP1, resulting in the reduction of its phosphorylation levels. This low steady state of WIP1 phopho-levels precisely stabilizes the protein at the time for ATM signaling termination [168]. Thus, the post-transcriptional regulation of WIP1 by phosphorylation constitutes another layer in the accurate regulation of the phosphatase activity during the damage response. This regulatory mechanism ensures the precise activation of WIP1 only when the DNA lesion has been fixed, avoiding the premature deactivation of the DNA damage checkpoint.

## 5. Concluding Remarks

Over the last few years, there has been the realization that the regulation of the DDR is vital to ensure genome stability. Consequently, there has been a burst in the number of studies headed to determine the fundamental principles behind the pathway and the regulatory mechanisms behind this biological process. While phosphorylation of multiple targets by DDR-kinases and their implications in the execution of the response is a well-known process, less is known about the importance of the DDR-phosphatases, their targets in the process, and their implication in genome integrity. Fortunately, we have recently started to focus our attention in the distinctive role of each DDR-related phosphatase in the control of the DNA repair process and their implications in the correct execution of the DDR. These discoveries have put forward a new concept in the field, where protein phosphatases are not passive members of the DDR but highly regulated enzymes with fundamental implications in the orchestration of the repair pathway. Taking into account the close relationship between DDR efficiency and cell transformation, it is intuitive to think that protein phosphatases could act as tumor suppressors. Supporting this notion, numerous carcinoma cells present an altered expression pattern and mutations in the sequence of several protein phosphatases [169,170,171]. Interestingly, protein phosphatases have been considered as good candidates for cancer therapy. In this regard, some experiments have pointed out the use of phosphatase-activating drugs in order to antagonize cancer development and progression. Moreover, the chemical reactivation of some phosphatases eradicates cancer cells while spanning normal cells. These observations make protein phosphatases attractive therapeutic targets for developing new treatment protocols. Therefore, due to the potential links between DDR-phosphatases and cancer, it is of vital relevance to comprehend how these proteins are regulated in response to a DNA lesion, their targets in the process, and their precise function in the regulation of the damage response. Understanding these fundamental questions will give us an enormous advantage in the knowledge of the DNA repair process and its implication in genome stability. 

## Figures and Tables

**Figure 1 ijms-21-00446-f001:**
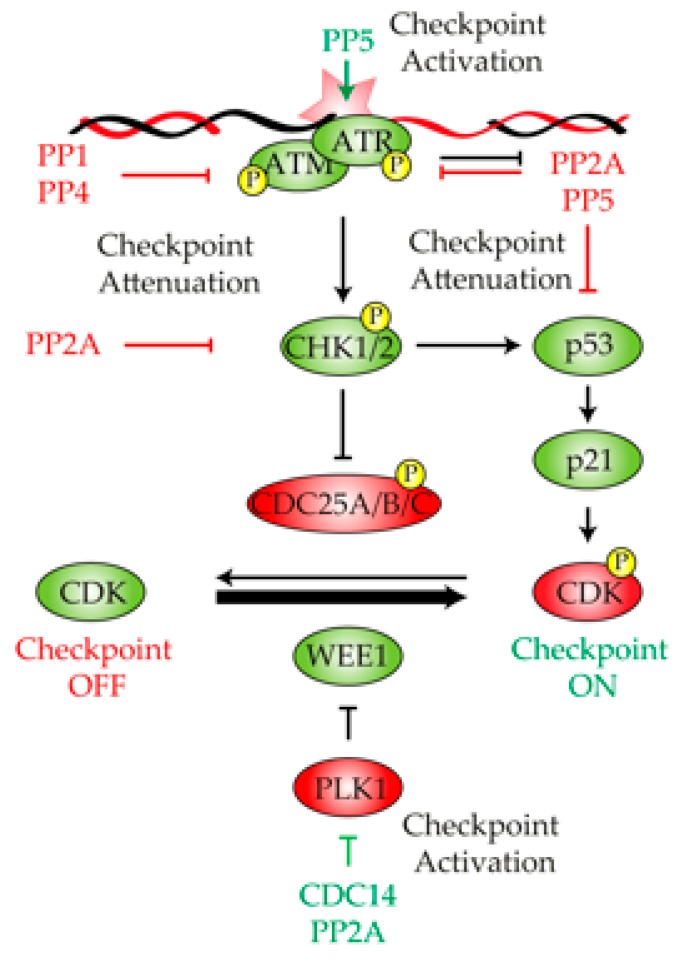
The diagram summarizes relevant information obtained from different model organisms, describing the role of protein phosphatases in the DNA damage checkpoint regulation. In response to DNA damage, phosphoprotein phosphatase 5 (PP5) collaborates in the activation of the DNA damage checkpoint by stimulating ATM/ATR activity, a process that triggers a phosphorylation cascade that end ups with the inhibition of the cyclin-dependent kinase (CDK). PP1, PP4 and PP2A restrain checkpoint activity by dephosphorylating ATM, ATR and p53, thus buffering the intensity of the response. CDC14 collaborates in the activation of the DNA damage checkpoint by restraining the negative effect that PLK1 exerts over the CDK inhibitor WEE1. Green and red indicate active or inactive, respectively.

**Figure 2 ijms-21-00446-f002:**
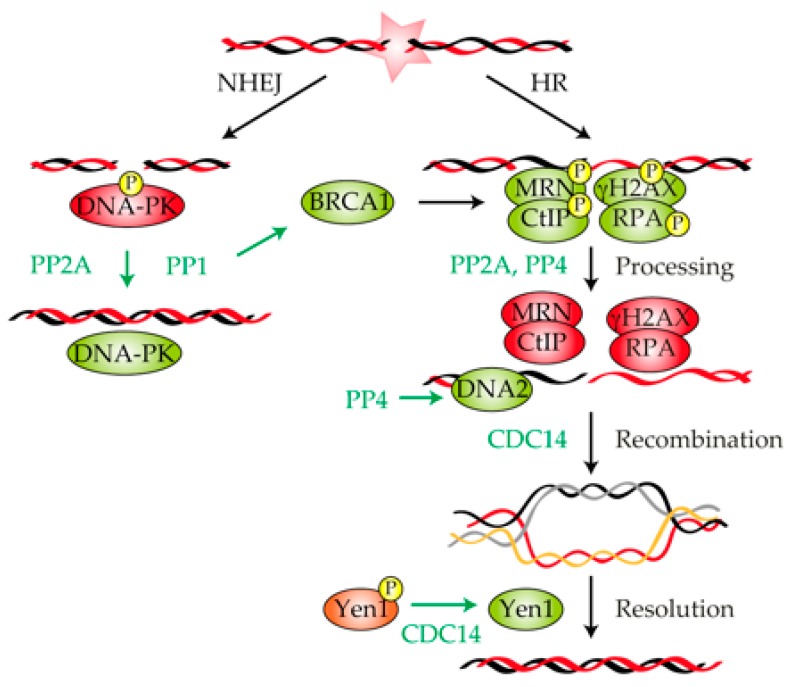
Involvement of protein phosphatases in the repair of a DNA lesion. Schematic representation showing the repair of a double-strand break (DSB) by non-homologous end joining (NHEJ) and homologous recombination (HR). Key factors involved in the execution of each repair pathway are shown. PP2A and PP1 stimulate NHEJ by dephosphorylating and activating DNA-protein kinase (PK). The dephosphorylation of MRE11–RAD50–NBS1 (MRN), γ-H2AX, CtIP and RPA by PP2A and PP4 is required for the processing and repair of the DNA lesion by HR. PP4 participates in HR by facilitating DNA2 accessibility to the DSB vicinity, thus enhancing DNA end resection. CDC14 promotes both recombinational DNA repair and resolution of recombinant intermediates. Green and red represent active or inactive, respectively.

**Figure 3 ijms-21-00446-f003:**
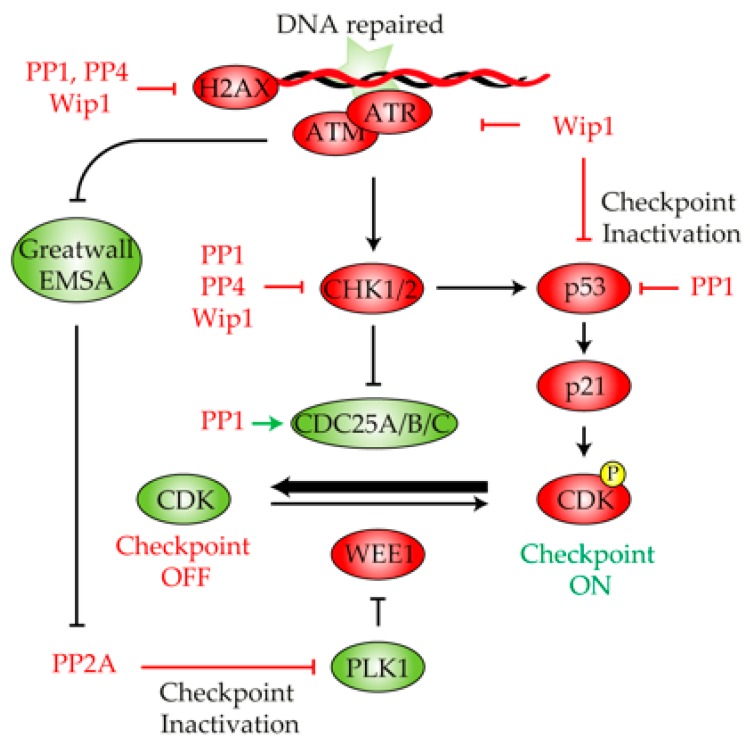
Role of protein phosphatases in the DNA damage checkpoint deactivation. Once the DNA lesion has been fixed, multiple phosphatases cooperate in the restoration of the phosphorylation state of multiple components of the DNA damage checkpoint pathway, thus stimulating cell cycle reactivation and recovery. PP1, PP4 and WIP1 dephosphorylate γ-H2AX and CHK1/2 to restrain the DNA damage signaling pathway. Wip1 also contributes to checkpoint deactivation by acting over ATM, ATR and p53. PP1 inhibits p53 and stimulates CDC25 activity, thus biasing the DNA damage checkpoint to an inactive state. The reactivation of Greatwall counteracts the PP2A inhibition of PLK1, thus inhibiting WEE1 activity. Green and red indicate active or inactive, respectively.

**Table 1 ijms-21-00446-t001:** A global overview of the role of protein phosphatases in the DNA damage response (DDR). The table contains information regarding the functions and targets of different protein phosphatases that are implicated in the regulation of the DNA damage response.

Function	PPase	Target
**DNA damage checkpoint activation**	PP1	hATM, hH2AX, h53BP1, hRPA, hRAD51, hCHK1
	PP2A	hATR, hATM, hDNA–PK, hCHK1, hCHK2, hPLK1, hP53
	PP4	scMec1
	PP5	hATM, hATR, hDNA–PK, mP53, mCHK1
	CDC14	spCds1, hCDH1
**DNA repair**	PP1	hBRCA1
	PP2A	hDNA–PK, hRPA, hH2AX, hATM, hCHK2
	PP4	ScRad53, ScH2A, hRPA, hH2AX
	CDC14	scSpc110, scYen1
**DNA damage checkpoint deactivation**	PP1	xCDC25C, hP53, caRad53, scH2A, ScRad53, spChk1
	PP2A	hPLK1
	PP4	scRad53, scH2A, hH2AX, dH2AX
	Wip1	hP53, hCHK1, hCHK2, hH2AX, mATM

Sc, Saccharomyces cerevisiae; Sp, Schizosaccharomyces pombe; Ca, Candida albicans; d, Drosophila melanogaster; x, Xenopus laevis; m, mice; h, human.

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
