# Peer review of "Cell Cycle and DNA Repair Regulation in the Damage Response: Protein Phosphatases Take Over the Reins"

_ijms, 2020, doi:10.3390/ijms21020446_

Round 1

Reviewer 1 Report

The DNA damage response is an intricate signaling network of proteins that respond to exogenous and endogenous stresses. The DDR is regulated primarily by three apical kinases, although there are functions for other post-translational modifications. In this review, the authors attempt to present a comprehensive view of the roles of protein phosphatases in the DDR.  I think that this topic will be of interest to a broad audience.

Major Comments:

The review was hard to read- I think that organization might be changed to better orient the reader. For example, it is a really hard distinction between checkpoint modulation and repair (since CHK1/2 have functions in both). Thus, I think it might be better if the authors organized the review by phosphatases, and then discussed the different functions under each phosphatase. The figures, while helpful are very broad; while the text is very detailed. It might help to have the figures focus on specific substrates and how that impacts the functions. For example, YEN1/RPA phosphorylation and dephosphorylation (and how that impacts functions) as cartoons will be much more useful than the line diagrams. A table of the phosphatases and their targets, along with the effect (attenuation versus enhancement of the DDR/cell cycle phase transitions) would be very useful and informative.

Other comments:

Line 107: the authors claim that several DNA repair factors have multiple sites that are phosphorylated and dephosphorylated concurrently. However, they do not cite any examples and there are no references. This has to be rectified Legend for figure 1: This legend needs work. The current text does not really describe all the players and their functions Lines 176-180: references needed Line 221: the authors introduce the subunits of PP2A without any background. Also this sentence involves three or four proteins that would greatly benefit from a cartoon to better make the point. Line 262: the authors discuss experiments in a particular strain of yeast (mec1-100) without providing information/background to the reader on what that strain is. Section 2.4: This section was very confusing. The authors (in line 300) make the point that PP5 enhances DNA-PK activity rather than ATM. In the next paragraph, they then describe other phenotypes observed on PP5 loss (Line 305). However, it is not apparent how those phenotypes connect with PP5 functions in modulating DNA-PK. Line 362-365: references needed. Lines 436-442: This paragraph was very confusing. The authors should clarify what they are talking about in this section. Section 3.3: The authors jump between different model systems that is sometimes very confusing, and while it is normal to try to integrate findings from diverse organisms, they should clarify what studies they are referring to in each section. Line 352: Reference?

Author Response

Major concerns

1) Reviewer proposes change the organization of the manuscript. He/She suggests organizing the review by phosphatase instead of DDR stages.

We agreed with the reviewer that the dissection of each DDR-related phosphatase along the different stages of the damage response might result a bit complicated. As commented by the reviewer, this is probably due to the overlapping roles that different DDR-related PPases presents in the damage response. However, we decided to organize the review in this way due to two different reasons. First, there are already a great number of reviews in the bibliography that have used this PPase classification. Indeed, our group published at the beginning of this year a review about PPases in the DDR by using this type of organization. Second, we believe that by focusing on the role of DDR-related PPases along the different stages of the DDR, we provide a novel point of view that might be more convenience for researchers working on specific processes of the DNA damage response.

2) Reviewer suggests modify figures to make them more intuitive and easy to follow by readers. He/She proposes focussing on PPases´s targets and functions instead of line diagrams.

As suggested by the reviewer, we have now completely changed the structure of figures 1, 2 and 3. The new figures contain detailed information regarding the targets of the PPases during checkpoint activation, repair and checkpoint silencing. We have also included information regarding the function of each dephosphorylation event during the processes. Moreover, and as suggested also by reviewer#2, figures have been converted to colourful. In this regard we have incorporated a colour code (green and red for active or inactive, respectively) in order to facilitate the interpretation of the different pathways of the DDR. I admit that the new way to represent the information enclosed in the text is more intuitive and easy to follow.

3) Reviewer advises incorporating a table containing the roles and targets of each DDR-related phosphatase.

As proposed by the reviewer, we have included a table containing the targets and the role of each protein phosphatase in DNA damage checkpoint activation, DNA repair and cell cycle re-entry.

Other comments

 1) Previous line 107. Reviewer suggests showing examples of kinases and phosphatases switches working in the DDR.

A new phrase denoting this observation has been included. We have also referenced a few examples of kinases/phosphatases molecular switches in the DDR.

2) Reviewer suggests improving figure legend 1 by describing key factors and function of DDR components.

We have been more explicit with the text enclosed in figure legend 1. A brief description of the different phosphatases´s targets and functions has been included. The same approach has also been extended to figure legends 2 and 3.

3) Reviewer advises include a reference in previous line 176-180.

We have included a reference accordingly to the reviewer suggestion.

4) Previous line 221. Reviewer suggests discussing about the different subunits of the PP2A holoenzyme at the beginning of the section.

We have incorporated a new paragraph describing the composition of the PP2A complex. We have also incorporating two references in the text containing relevant information about PP2A structure and composition.

5) Previous Line 262. Reviewer suggests clarify the nature of the mec1-100 allele.

We have included a brief description of the mec1-100 mutation in the text.

6) Previous line 300. Reviewer claims to clarify the paragraph describing PP5 role in the control of the DDR by targeting ATM/DNA-PK.

We are totally agreed with the reviewer that the information included in this paragraph was a little disorganised. We have now mended the problem by separating both ATM-ATR and DNA-PK dependent regulation by PP5 in the text.

7) Reviewer suggests adding reference to previous lines 362-365.

We have now inserted two new references at the introduction of section 3. The first one includes information about the resection process. The second reference has been incorporated to specifically dissect the homologous recombination pathway.

8) Previous lines 436-442. Reviewer suggests clarify the paragraph.

We have rephrased the paragraph to make it more accessible to readers. Briefly, we have leave clear that PP2A is involved in DNA repair by controlling the steady state phosphorylation of checkpoint proteins. We have also speculate whether this function is a direct effect of the phosphatase in controlling DDR targets or a consequence of its function in cell cycle progression.

9) Section 3.3. Reviewer suggests integrating the findings from the diverse organisms used in this section.

As suggested by the reviewer, we have specifically noted in the text the model organism used in every statement of the section. We have also separated the information regarding the budding yeast (top part of the section) from the high eukaryotes data (bottom part of the section).

10) Previous line 352. Reviewer suggests including a reference.

As suggested by the reviewer, we have added a reference when discussing the types of DNA repair.  

Reviewer 2 Report

The review „Cell cycle and DNA repair regulation in the damage response: protein phosphatases take over the reins“ by A. Campos & A. Clemente-Blanco summarizes in detail recent findings about protein phosphatases and their role in DNA damage response and maintenance of genome integrity. Authors provide thorough description of the role of known protein phosphatases in control of the DNA damage checkpoints, in repair of DNA lesions and in DNA damage checkpoint deactivation and cell cycle recovery.

In my opinion, the review will attract broad range of audience, including those studying the mechanisms of DNA damage response and protection of genome integrity. In my opinion it might be published in its current version.

Minor comment: It would probably be better if schemes were colourful.

Author Response

Reviewer#2

Minor comments

1) Reviewer suggests incorporating colourful schemes.

As proposed by the reviewer we have substituted previous black and white figures for colourful schemes.

Reviewer 3 Report

The review presented by the authors covers the recent development on the understanding of the role of different kinase and posphatase enzymes in regulating the complex response to DNA damages, including the temporary cycle blockage, the activation of the repair machinery, and the reinitialization of the cycle once the lesions have been removed. 

In particular it has been shown how the different proteins may assume different role in the regulation of the mechanisms and how this could influence the complex crosstalks taking place in the cells. 

I think the review is welcomed providing a good overview of an important question with strong potential influence in advancing the understanding of DNA repair mechanisms, i.e. an extremely fundamental process with great significance also for cancer therapy. Hence I think its publication would be welcomed. However some revision are necessary prior to its acceptance. 

First of all the abstract and the introduction are quite poorly written and will require a significant proof reading to remove badly constructed sentences or even some totally inappropriate phrasing. Some examples including the lesions "insulting" the repair process. 

Itr would be nice if the introduction when dealing with the different repair mechanisms, BER, NER, HR, etc... the authors could also recall the different lesions that are process by those pathways. 

I understand that the review deals more with the regulatory interactions but if some structures of the different enzymes and of the putative or confirmed contact region are present I think they should be reported to further strenghten the impact of the review. This is particularly  true for a journal mostly dealing with molecular sciences. 

Also a global analysis of the role and specificity, if any, of the ddifferent phosphatase in regulating the response to different class of D?NA lesions could be appropriate. 

The affirmation made in the conclusion concerning carcinoma cells presenting alteration in the gene expression of phosphatases, as well as the interest for these latter as anticancer targets should be supported by appropriate references. 

Finally, and as a minor point, I would not put atmospheric pollution on the same footing of the other stress sources indeed strictly speaking one could think of pollution as an accumulation of the other stress factors such as hydrocarbon, radiations, etc.... 

Author Response

Reviewer#3

General comments

1) Reviewer suggests revise the abstract and introduction to avoid badly constructed sentences and inappropriate phrasing.

We have now revised the text looking for grammatical errors and non-canonical English expressions.

2) Reviewer recommends recall in the different lesions that each type of repair pathway resolve.

We have including this information in the introduction when discussing about the different repair system used by the cell.

3) Reviewer suggests discussing about each phosphatase structure and the putative or confirmed contact regions with their substrates/regulators.

Since this is not the topic of the revision, we have only included a brief description about each phosphatase structure at the beginning of each section. We have also concisely discussed about possible interactive motives of each phosphatase core subunit with substrates/regulators. At all cases, we have supplemented the text with updated references about the topic.

4) Reviewer suggests including references in the discussion section related to the alteration in phosphatase expression.

We have included a few references accordingly reviewers suggestion.

5) Reviewer advises to make an analysis of the role and specificity of DDR-phosphatases in response to different classes of DNA lesions.

We agree with the reviewer that different sources of DNA damage could attain for different protein phosphatases requirements. Indeed, the previous version of the manuscript already contained relevant information regarding the type of DNA damage infringed on each experimental condition assayed. We have carefully revised the manuscript and improved this notion in the new version of the review.

6) Reviewer suggests eliminating atmospheric pollution as an example of stress source.

We have removed the sentence from the text.

Round 2

Reviewer 1 Report

The authors have addressed most of my concerns. The new figures are well done, and the manuscript is significantly improved. I think that the review can be accepted in its present form.

Reviewer 3 Report

The authors have answered to my main concerns. There are still some typos or grammar errors that could be safely corrected at proof reading stage. 

Hence I suggest acceptation of the present manuscript